# ON THE STATISTICAL AND INFORMATION THEORETICAL CHARACTERISTICS OF DNN REPRESENTATIONS

## ABSTRACT

It has been common to argue or imply that a regularizer can be used to alter a statistical property of a hidden layer's representation and thus improve generalization or performance of deep networks. For instance, dropout has been known to improve performance by reducing co-adaptation, and representational sparsity has been argued as a good characteristic because many data-generation processes have only a small number of factors that are independent. In this work, we analytically and empirically investigate the popular characteristics of learned representations, including correlation, sparsity, dead unit, rank, and mutual information, and disprove many of the *conventional wisdom*. We first show that infinitely many Identical Output Networks (IONs) can be constructed for any deep network with a linear layer, where any invertible affine transformation can be applied to alter the layer's representation characteristics. The existence of ION proves that the correlation characteristics of representation can be either low or high for a well-performing network. Extensions to ReLU layers are provided, too. Then, we consider sparsity, dead unit, and rank to show that only loose relationships exist among the three characteristics. It is shown that a higher sparsity or additional dead units do not imply a better or worse performance when the rank of representation is fixed. We also develop a rank regularizer and show that neither representation sparsity nor lower rank is helpful for improving performance even when the data-generation process has only a small number of independent factors. Mutual information $I(\mathbf{z}_l; \mathbf{x})$ and $I(\mathbf{z}_l; \mathbf{y})$ are investigated as well, and we show that regularizers can affect $I(\mathbf{z}_l; \mathbf{x})$ and thus indirectly influence the performance. Finally, we explain how a rich set of regularizers can be used as a powerful tool for performance tuning.

## 1 INTRODUCTION

A learned representation can significantly affect the performance of deep networks, and the representation's distributed and deep natures are the essential elements for the success of deep learning (Bengio et al., 2013). As a consequence, deep networks have a greater *expressiveness* compared to the other machine learning algorithms (Hinton et al., 1986) or shallow networks (Montufar et al., 2014; Telgarsky, 2015; Eldan & Shamir, 2016; Raghu et al., 2016). Besides the distributed and deep natures that have been intensively studied, a hidden layer's representation characteristics are considered to be important as well. Nonetheless, a relatively smaller number of studies have been completed on the topic, and the goal of this work is to understand the representation characteristics better. Therefore, *the meaning of representation in this work is restricted to the activation vector of a single hidden layer* and a *unit* refers to a neuron of the hidden layer.

A few previous studies considered manipulating *statistical characteristics* of representations such as reducing covariance among hidden units (Cogswell et al., 2015; Xiong et al., 2016), encouraging representational sparsity (Glorot et al., 2011), or forcing parsimonious representations via clustering (Liao et al., 2016). In some of the similar works, a popular argument has been that the representation regularization reduces the generalization error via altering a representation characteristic. This argument, however, has not been rigorously studied. Another popular argument has been the reduction of effective capacity via regularization. This argument has been recently disproved by Zhang et al. (2016) where they empirically show that explicit regularization methods like L2 weight decay and dropout cannot sufficiently limit the effective capacity of deep networks.

Table 1: Representation characteristics.

| Characteristic | Symbol | Expression |
|---|---|---|
| ACTIVATION AMPLITUDE | $\lvert z \rvert$ | $\mathbb{E}_i[\lvert \mathbf{z}_{l,i} \rvert]$ |
| COVARIANCE | $\bar{c}$ | $\mathbb{E}_{i \neq j}[c_{i,j}]$, where $c_{i,j} \triangleq \{\mathbf{C}_l\}_{i,j} = \mathbb{E}[(\mathbf{z}_{l,i} - \mu_{z_{l,i}})(\mathbf{z}_{l,j} - \mu_{z_{l,i}})]$ |
| CORRELATION | $\bar{\rho}$ | $\mathbb{E}_{i \neq j}[\rho_{i,j}]$, where $\rho_{i,j} \triangleq \{\mathbf{C}_l\}_{i,j}/\sigma_{z_{l,i}}\sigma_{z_{l,j}} = \mathbb{E}[(\mathbf{z}_{l,i} - \mu_{z_{l,i}})(\mathbf{z}_{l,j} - \mu_{z_{l,i}})]/\sigma_{z_{l,i}}\sigma_{z_{l,j}}$ |
| SPARSITY | $P_s$ | $\mathbb{E}_{i,n}[\mathbb{1}(z_{l,i}^n)]$, where $\mathbb{1}$ is an indicator function whose output is 1 only when $z_{l,i}^n = 0$ |
| DEAD UNIT | $P_d$ | $\mathbb{E}_i[\mathbb{1}(z_{l,i})]$, where $\mathbb{1}$ is an indicator function whose output is 1 only when $z_{l,i}^n = 0$ for all $n = 1, .., N$ |
| RANK | $r$ | $rank(\mathbf{C}_l)$; numerical evaluations are approximated as the stable rank $\lVert \mathbf{C}_l \rVert_F^2 / \lVert \mathbf{C}_l \rVert_2^2$ |
| MUTUAL INFORMATION | $I_{\mathbf{x}}$ | $I(\mathbf{z}_l; \mathbf{x})$ |
| MUTUAL INFORMATION | $I_{\mathbf{y}}$ | $I(\mathbf{z}_l; \mathbf{y})$ |

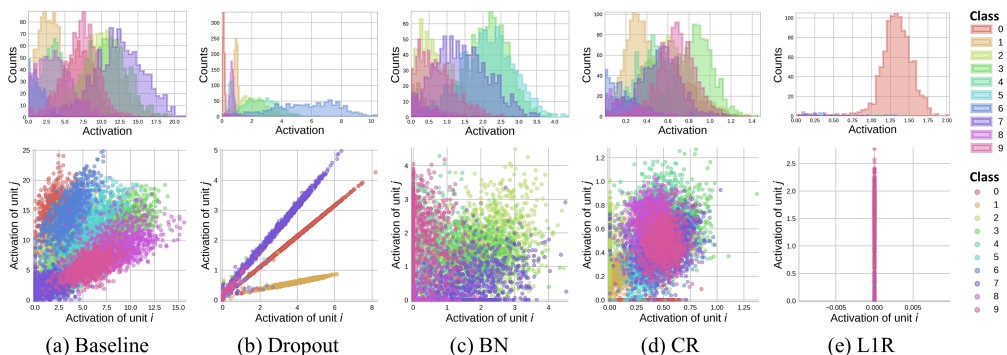

| (a) Baseline | (b) Dropout | (c) BN | (d) CR | (e) L1R |
|---|---|---|---|---|

Figure 1: Visualization of the learned representations for a 6-layer MLP trained with MNIST dataset. A single unit's activation histogram (upper plots) and two randomly chosen units' activation scatter plots (lower plots) are shown for the fifth layer's representation, where each color corresponds to a different class. The plots were generated using 10,000 test samples of MNIST dataset. **(Upper)** It can be seen that the baseline has a large class-wise variance and inter-class overlaps, and BN and CR (originally known as DeCov (Cogswell et al., 2015)) show similar properties. Dropout looks completely different where activation values are more spread out for the active classes. L1R (L1 Representation regularizer) typically allow only one or two classes to be activated per unit. **(Lower)** While the baseline shows modest linearity, the others show quite different representation characteristics depending on the choice of regularizer. Dropout shows an extremely high class-wise correlation, but BN shows very low correlation. CR shows almost no correlation. Since L1R increases sparsity on representation, a class is activated over at most one of the two randomly chosen units.

Since a novel information-theoretic analysis method was proposed for deep learning (Tishby & Zaslavsky, 2015; Shwartz-Ziv & Tishby, 2017), *information theoretic* characteristics of representation have become an important research topic. In their work, mutual information $I(\mathbf{z}_l; \mathbf{x})$ and $I(\mathbf{z}_l; \mathbf{y})$ are used to address the learning dynamics and generalization of deep learning, where $\mathbf{z}_l$ is the hidden layer $l$'s representation, $\mathbf{x}$ is the input, and $\mathbf{y}$ is the output. It is further discussed that a good representation is the one that contains a minimal amount of information from the input while containing a sufficient amount of information from the output. In Achille & Soatto (2017), the Information Bottleneck Lagrangian (Tishby et al., 2000) is decomposed into the sum of a cross-entropy term and a regularization term. The regularization term turns out to be $I(\mathbf{z}_l; \mathbf{x})$ that needs to be minimized. Some of the recent works will be additionally addressed in Section 5.

## 2 REPRESENTATION CHARACTERISTICS

In this section, we briefly address the most popular statistical characteristics and information theoretic characteristics of representations. Consider a neural network $\mathcal{N}_{\mathcal{A}}$ whose architecture $\mathcal{A}$ is fixed and the weights for the $l^{\text{th}}$ layer are given by $\{\mathbf{W}_l\}$ and $\{\mathbf{b}_l\}$ after training. Notation-wise, we simply write $\mathcal{N}_{\mathcal{A}} = (\mathbf{W}, \mathbf{b})$ to define a network and $\mathbf{y}$ or $\mathcal{N}_{\mathcal{A}}(\mathbf{x})$ to refer to its deterministic output for a given input $\mathbf{x}$. The layer index $l$ is omitted when the meaning is obvious. The $l^{\text{th}}$ layer's activation vector for the given input $\mathbf{x}$ is noted as $\mathbf{z}_l(\mathbf{x})$ or simply $\mathbf{z}_l$, and the $i^{\text{th}}$ element of $\mathbf{z}_l$ is noted as $z_{l,i}$. The mean, variance, and standard deviation of $z_{l,i}$ are defined as $\mu_{z_{l,i}}$, $v_{z_{l,i}}$, and $\sigma_{z_{l,i}}$, respectively. The covariance of $\mathbf{z}_l$ is defined as $\mathbf{C}_l$. Then, the basic representation characteristics can be summarized as in Table 1. Six of them are statistical characteristics, and the last two are information theoretic characteristics.

Previous studies on statistical characteristics are often based on regularizers. Srivastava et al. (2014) addresses dropout for preventing co-adaptation among hidden units by randomly putting zeros in a layer's activation vector. Ioffe & Szegedy (2015) explains batch normalization (BN) that reduces internal covariate shift via normalizing activations of each unit to speed up network training. Cogswell et al. (2015) suggest DeCov that utilizes a penalty loss function to reduce activation covariance among hidden units. Choi & Rhee (2018) considers extension to class-wise regularization and provides four representation regularizers. Among them, CR (Covariance Regularizer) is equivalent to DeCov, and we adopt the notation of CR. Glorot et al. (2011) explains L1 representation regularization, called L1R in this work, that applies L1 penalty on activations. These representation regularization methods have distinct effects on representation characteristics, and examples of the learned representations are shown in Figure 1. The representation regularizers are described in Appendix B, and a quantitative analysis of representation characteristics are provided in Appendix C.

Because the true distribution of data is not accessible, the numerical results in the following sections are evaluated using the empirical distribution of the test dataset. Then, the expectations in Table 1 are with respect to the empirical distribution. For instance, $\mathbf{C}_l$ is calculated as the covariance matrix of $N$ activation vectors $\{\mathbf{z}_l^1, ..., \mathbf{z}_l^N\}$ where $\mathbf{z}_l^n$ corresponds to the activation vector for the $n$'th test data sample, $\mathbf{x}^n$. Rank can be calculated by examining $\mathbf{C}_l$, but often there are tiny eigenvalues that hinder a proper assessment of the rank. Therefore, we evaluate *stable rank* instead, and it will be explained further in Section 4. Two information-theoretical characteristics, $I(\mathbf{z}_l; \mathbf{x})$ and $I(\mathbf{z}_l; \mathbf{y})$, are estimated using upper and lower bounds explained in Kolchinsky & Tracey (2017); Kolchinsky et al. (2017). Further details are provided in Section 5. ReLU is the only activation function that is considered in this work. When ReLU is used, ACTIVATION AMPLITUDE, COVARIANCE, and CORRELATION are calculated using only the positive activation values such that the numerical evaluations can provide meaningful insights on what is happening to the non-zero representation values.

## 3 SCALING, PERMUTATION, COVARIANCE, AND CORRELATION

After training is completed for a deep network $\mathcal{N}_\mathcal{A}$, the output of the network becomes a deterministic function of the input $\mathbf{x}$. Without an activation function, i.e. a linear layer, $\mathbf{z}_l = \mathbf{W}_l^T \mathbf{z}_{l-1} + \mathbf{b}_l$. When ReLU is applied to layer $l$, the activation vector becomes $\mathbf{z}_l = ReLU(\mathbf{W}_l^T \mathbf{z}_{l-1} + \mathbf{b}_l) = \max(\mathbf{W}_l^T \mathbf{z}_{l-1} + \mathbf{b}_l, 0)$. In this section, we investigate the most flexible affine transformation that can be applied to a layer's representation $\mathbf{z}_l$ without influencing the output $\mathcal{N}_\mathcal{A}(\mathbf{x})$ for any arbitrary input vector $\mathbf{x}$. While complicated transformations over multiple layers can be explored, we limit our focus to manipulating only the weights of layer $l$ and layer $l+1$ for the analytical tractability. Because scaling and permutation are well known results, covariance and correlation are the main focus of this section.

### 3.1 IDENTICAL OUTPUT NETWORK (ION)

We first consider a linear layer $l$. For a linear layer, it turns out that any affine transformation can be applied as long as the transformation does not cause an information loss.

**Theorem 1.** *(ION for a linear layer) For a deep network $\mathcal{N}_\mathcal{A} = (\mathbf{W}, \mathbf{b})$ whose layer $l$ is linear, there exists $\widetilde{\mathcal{N}}_\mathcal{A} = (\widetilde{\mathbf{W}}, \widetilde{\mathbf{b}})$ that satisfy the following conditions:*

$$\forall \mathbf{x}, \ \mathcal{N}_\mathcal{A}(\mathbf{x}) = \widetilde{\mathcal{N}}_\mathcal{A}(\mathbf{x}); \tag{1}$$

$$\forall \mathbf{x}, \ \widetilde{\mathbf{z}}_l = \mathbf{Q}(\mathbf{z}_l - \mathbf{m}), \tag{2}$$

*where $\mathbf{Q}$ is any nonsingular square matrix of a proper size and $\mathbf{m}$ is any vector of a proper size.*

The first condition says that the two networks generate identical outputs for any input $\mathbf{x}$. The second condition says that $\mathbf{z}_l$ can be affinely transformed using any nonsingular matrix $\mathbf{Q}$. The proof is straightforward and can be found in Appendix A.

While simple, Theorem 1 has significant implications on the representation characteristics of $\mathbf{z}_l$. Let's inspect covariance and correlation (normalized version of covariance) first. If $\mathcal{N}_\mathcal{A}$ is a network that is globally optimal for a task and has at least one linear layer $l$, then $\mathcal{N}_\mathcal{A}$'s covariance $\mathbf{C}_l$ can be whitened to have $\widetilde{\mathbf{C}}_l = \mathbf{I}$ by choosing $\mathbf{m}$ as the expected mean and $\mathbf{Q}$ as a whitening matrix. The

resulting network $\widetilde{\mathcal{N}}_{\mathcal{A}}$ will have zero correlation between any pair of units in layer $l$, but will be globally optimal, too. In fact, there are infinitely many globally optimal networks with different covariance characteristics, and one can easily construct an ION with an arbitrary covariance matrix $\widetilde{\mathbf{C}}_l$ as long as its rank is the same as $\mathbf{C}_l$'s rank. With this result, it becomes unclear why one should pursue a lower correlation when training a deep network. Unless regularization for a low correlation somehow helps optimization to reach a better performing network, there seems to be no reason to pursue low (or high) correlation.

For dead neurons, a similar claim can be made. If globally optimal $\mathcal{N}_{\mathcal{A}}$ has no dead neurons in layer $l$ and $\mathbf{C}_l$ is not full rank, one can make an affine transformation to align the null spaces of $\mathbf{C}_l$ to some of the neurons. Then, the resulting network $\widetilde{\mathcal{N}}_{\mathcal{A}}$ will be still globally optimal, but with some dead neurons in layer $l$. For higher layers of classification tasks, typically the rank of $\mathbf{C}_l$ is close to the number of classes. For classification tasks with only 2~10 classes, it is possible to construct an ION that has as many dead neurons as the size of $\mathbf{C}_l$'s null space. This can be done without negatively affecting the performance, and the wisdom of 'reduce the number of dead neurons' becomes dubious.

For scaling and permutation, their influences are rather insignificant. As for the scaling that can affect activation amplitude, it often has no effect on the network's performance. For instance, scaled activation amplitude can affect the probability of classification tasks when softmax is in the last layer, but the class with the highest probability remains the same anyway. When representation regularizers are used, often activation amplitude is squashed to reduce the cost of representation penalty function but the network can still perform well. As reported in Choi & Rhee (2018), such an activation squashing can make covariance much smaller, but the effect is removed when correlation is calculated. As for the permutation, it is considered to be meaningless because the index number itself is not important.

Before discussing further, a similar result is developed for ReLU layers. The resulting $\mathbf{Q}$, however, is much more limited. The proof can be found in Appendix A as well.

**Theorem 2.** *(ION for a ReLU layer) For a deep network $\mathcal{N}_{\mathcal{A}} = (\mathbf{W}, \mathbf{b})$ whose activation function of layer $l$ is ReLU, there exists $\widetilde{\mathcal{N}}_{\mathcal{A}} = (\widetilde{\mathbf{W}}, \widetilde{\mathbf{b}})$ that satisfy the following conditions:*

$$\forall \mathbf{x}, \ \mathcal{N}_{\mathcal{A}}(\mathbf{x}) = \widetilde{\mathcal{N}}_{\mathcal{A}}(\mathbf{x}); \tag{3}$$

$$\forall \mathbf{x}, \ \widetilde{\mathbf{z}}_l = \mathbf{Q}\, \mathbf{z}_l, \tag{4}$$

*where $\mathbf{Q}$ is any permuted positive diagonal matrix of a proper size. Furthermore, it can be shown that any $\mathbf{Q}$ that satisfy the above two conditions must be a permuted positive diagonal matrix.*

Using a permuted positive diagonal matrix $\mathbf{Q}$, covariance can be affected by independently scaling activation amplitudes of layer $l$'s units. As explained above, such a scaling is canceled out when calculating correlation and therefore a linear transformation cannot affect correlation while keeping the output identical. There are a few possibilities for overcoming the limitations of Theorem 2, and they are discussed in the following subsection.

For rank and mutual information, the invertible affine transformation has no effect. They are discussed in the following sections.

## 3.2 POSSIBLE EXTENSIONS OF ION

We discuss three possible extensions of ReLU's ION.

**Insertion of a linear layer** One way to overcome the limitations of $\mathbf{Q}$ is to insert an extra linear layer near the target ReLU layer and to consider its implications. When representation characteristics are analyzed or interpreted, researchers do not care much about the layer's activation function, regularization, etc. The activation vector's representation characteristics are the essential components for understanding and assessing the representations. Therefore, one can apply the insights from Theorem 1 when the extra linear layer shows similar statistical properties as the ReLU layer. In Table 2, statistical properties of immediately neighboring ReLU and linear layers are compared. Compared to the representation characteristics of the first ReLU layer, the 5th ReLU layer and the inserted 6th linear layer show very similar characteristics. Then the correlation and dead unit characteristics are

Table 2: Comparison of statistical characteristics for linear and ReLU layers. A 7-layer MLP was used with MNIST dataset, and only the sixth layer was linear, and the others were ReLU layers. Statistical characteristics of the first layer (ReLU), fifth layer (ReLU), and sixth layer (linear) are compared. It can be seen that representation characteristics of fifth and sixth are very similar because they are both located in the upper part of the network. Note that the characteristics of the sixth layer were calculated only using positive activation values for fair comparison.

| Regularizer | ACTIVATION AMPLITUDE | COVARIANCE | CORRELATION | SPARSITY | DEAD UNIT | RANK |
|---|---|---|---|---|---|---|
| Baseline (1st) | 1.16 | 0.08 | 0.14 | 0.00 | 0.00 | 2.22 |
| Baseline (5th) | 2.22 | 0.45 | 0.25 | 0.32 | 0.07 | 2.27 |
| Baseline (6th) | 2.75 | 0.62 | 0.25 | 0.47 | 0.00 | 2.78 |
| Dropout (1st) | 0.76 | 0.04 | 0.27 | 0.86 | 0.00 | 3.50 |
| Dropout (5th) | 2.19 | 1.23 | 0.70 | 0.53 | 0.00 | 2.19 |
| Dropout (6th) | 1.85 | 0.97 | 0.53 | 0.48 | 0.01 | 1.52 |
| BN (1st) | 0.80 | 0.03 | 0.10 | 0.49 | 0.00 | 4.07 |
| BN (5th) | 1.01 | 0.10 | 0.19 | 0.51 | 0.00 | 4.59 |
| BN (6th) | 1.39 | 0.19 | 0.23 | 0.48 | 0.00 | 4.26 |

not so important as the consequence of Theorem 1, and the same might be conjectured for the 5th ReLU layer.

**Non-affine transformations over multiple layers** In the ION derivations, we have considered only an affine transformation applied to the layers $l$ and $l+1$ only. If we remove the constraints and borrow the results from expressivity of DNN and universal approximation theorem, it might be possible to derive more powerful and general results. In the extreme case, one can divide a deep network $\mathcal{N}_{\mathcal{A}}$ into two parts: $\mathcal{N}_{\mathcal{A}_{\text{lower}}}$ and $\mathcal{N}_{\mathcal{A}_{\text{upper}}}$. Then, $\mathcal{N}_{\mathcal{A}_{\text{lower}}}(\mathbf{x}) = \mathbf{z}_l$ and $\mathcal{N}_{\mathcal{A}}(\mathbf{x}) = \mathcal{N}_{\mathcal{A}_{\text{upper}}}(\mathcal{N}_{\mathcal{A}_{\text{lower}}}(\mathbf{x}))$. In theory, there exist $\widetilde{\mathcal{N}}_{\mathcal{A}_{\text{lower}}}$ and $\widetilde{\mathcal{N}}_{\mathcal{A}_{\text{upper}}}$ that can result in $\mathcal{N}_{\mathcal{A}}(\mathbf{x}) = \widetilde{\mathcal{N}}_{\mathcal{A}_{\text{upper}}}(\widetilde{\mathcal{N}}_{\mathcal{A}_{\text{lower}}}(\mathbf{x}))$ while allowing $\widetilde{\mathbf{z}}_l$ to have a completely different characteristics compared to $\mathbf{z}_l$. Such $\widetilde{\mathcal{N}}_{\mathcal{A}_{\text{lower}}}$ and $\widetilde{\mathcal{N}}_{\mathcal{A}_{\text{upper}}}$, however, might be infeasibly large or fail to learn in the way we desire. Therefore, it might be more practical to consider a reasonable extension of Theorem 1 and Theorem 2.

**Comparable Performance Network (CPN)** According to Theorem 2, only permuted positive diagonal matrices can form IONs. If we ignore the result and apply an affine transformation in the same way as in the ION of a linear layer, the resulting network $\widetilde{\mathcal{N}}_{\mathcal{A}}$ will not form an ION but instead we might be able to find a Comparable Performance Network (CPN) that achieves a comparable performance while showing different representation characteristics. We tried this brute-force method, and two sample results along with the baseline and a positive diagonal matrix case are shown in Table 3. In the first row where $\mathbf{Q}$ is identity, the values are for the original network $\mathcal{N}_{\mathcal{A}}$. In the next row, 'Random positive diagonal', uniformly random values between 0 and 1 ($U(0, 1)$) were used as the diagonal values. Note that this choice of $\mathbf{Q}$ satisfies Theorem 2, and therefore the error performance remains the same while affecting activation amplitude and covariance only. In the 'Ones with random positive diagonal', $\mathbf{Q}$ was chosen as a matrix of ones with its diagonals replaced with random values chosen from $U(0, 1)$. We randomly generated 100 of such random matrices and selected the one that resulted in a higher correlation while showing a comparable performance. Despite of the very high correlation of 0.80, the selected network $\widetilde{\mathcal{N}}_{\mathcal{A}}$ can perform comparably well. In the last row, we applied a whitening filter where $\mathbf{Q}$ and $\mathbf{m}$ were calculated while ignoring ReLU. The resulting network does not end up with zero correlation because the whitening is not perfect in the presence of ReLU. But the correlation is considerably reduced to 0.09 from 0.28 while achieving a slightly worse error rate of 5.48%.

To find the examples in Table 3, all we had to do was to construct a meaningful matrix $\mathbf{Q}$ or to try 100 random matrices and choose one. The fact that it is almost painless job to find a CPN also implies that the relevant representation characteristics might not be essential for achieving high performance.

## 4 SPARSITY, DEAD UNIT, AND RANK

Sparsity and dead unit have been considered as important representation characteristics. Rank of $\mathbf{C}_l$, however, has received much less attention so far. In this section, we investigate the three and show that rank might be the most fundamental characteristics.

Table 3: Statistical characteristics of representations transformed by CPNs. The original network is 6-layer MLP on the MNIST dataset, and the 5th layer representations were transformed. To improve the performance, the weights to the output layer were fine tuned after applying **Q**.

| Q matrix | Error (%) | ACTIVATION AMPLITUDE | COVARIANCE | CORRELATION | SPARSITY | DEAD UNIT |
|---|---|---|---|---|---|---|
| Identity | 2.54 | 6.79 | 4.29 | 0.28 | 0.36 | 0.13 |
| Random positive diagonal | 2.54 | **3.37** | **1.04** | 0.28 | 0.36 | 0.13 |
| Ones with random positive diagonal | 2.76 | 158.88 | 723.55 | **0.80** | **0.00** | **0.00** |
| Whitening | 5.48 | 1.22 | 0.96 | **0.09** | **0.49** | **0.02** |

## 4.1 ANALYTICAL RELATIONSHIP

In Table 1, sparsity is defined as $P_s = \mathbb{E}_{i,n}[\mathbb{1}(z_{l,i}^n)]$. This can be interpreted as the probability of $z_{l,i}^n$ (unit $i$'s activation for $n$'th test sample $\mathbf{x}^n$) being zero, because $\mathbb{1}(z_{l,i}^n) = 1$ when $z_{l,i}^n = 0$ and $\mathbb{1}(z_{l,i}^n) = 0$ when $z_{l,i}^n \neq 0$. Similarly, dead unit is defined as $P_d = \mathbb{E}_i[\mathbb{1}(z_{l,i})]$ and it can be interpreted as the probability of $z_{l,i}$ (unit $i$'s activation) being always zero or at least for all $M$ test samples. Because $\mathbb{1}(z_{l,i}) \leq \mathbb{1}(z_{l,i}^n)$ for any pair of $(i,n)$, $P_d \leq P_s$ can be shown by taking expectations on both sides. The rank $r$ in Table 1 is defined as the rank of $\mathbf{C}_l$. For layer $l$ with $M$ units, this means that $r$ out of $M$ linearly independent dimensions are used by the codewords $\{\mathbf{z}_l^1, ..., \mathbf{z}_l^N\}$ and that the other $M - r$ dimensions form a null space of $\mathbf{C}_l$. When dead units are considered, $MP_d$ units need to be constant zero by the definition of the dead unit and it implies that at least $MP_d$ dimensions need to be included in the null space. Therefore, $MP_d \leq M - r$. These results can be summarized as below.

$$P_d \leq P_s \tag{5}$$
$$MP_d \leq M - r \tag{6}$$

Between sparsity $P_s$ and rank $r$, there is no clear relationship. The codeword $\mathbf{z}_l^n$ for a test sample $\mathbf{x}^n$ can be very sparse, and yet the set of codewords $\{\mathbf{z}_l^1, ..., \mathbf{z}_l^N\}$ collectively might use all of the $M$ dimensions. Conversely, rank $r$ can be very small and yet $P_s$ can also be very small when $\{z_{l,1}^n, ..., z_{l,M}^n\}$ are strongly correlated and the basis vectors are not sparse over the $M$ units.

From the viewpoint of signal processing or information theory, sparsity is a property that is related to individual signals or individual codewords while rank is a property that is related to the total number of dimensions used by the set of signals or the entire codebook. Therefore, sparsity is not directly responsible for the efficiency of the signals or codebook while rank is directly responsible for the efficiency. From the viewpoint of deep learning, rank can be associated with the maximum number of latent factors that are independent. As for the dead unit, we know from equation 6 that it is upper bounded as a function of rank. When the bound is met, the value of $P_d$ is merely an artifact of how the representation vectors $\{\mathbf{z}_l\}$ are aligned with the eigenvectors of $\mathbf{C}_l$. If each neuron is aligned to an eigenmode of $\mathbf{C}_l$, then $P_d = 1 - r/M$ will be achieved. From our experience, however, such a perfect alignment never happens when using the backpropagation based learning process. This has been true even when L1R or other advanced representation regularizers were applied. ION, however, can easily meet the requirement for a linear layer.

Motivated by the above discussion, we have designed a rank regularizer and examined common wisdom that says 'most of the data generation processes have a small number of independent factors and therefore increasing sparsity of representation can be helpful.' For instance, see Bengio et al. (2013). We first explain the design of rank regularizer.

## 4.2 RANK REGULARIZER

In deep learning, a low-rank approximation of convolutional filters (Jaderberg et al., 2014; Lebedev et al., 2014; Tai et al., 2015) and weight matrices (Xue et al., 2013; 2014; Nakkiran et al., 2015; Masana et al., 2017) has been widely used for network compression and fast network training. Some of the works applied a singular value decomposition to weight matrices after network training ends but not to representations. In this work, as L1 representation regularizer was designed to encourage a higher sparsity by adding a penalty loss term $\Omega_{L1R} = \sum_n \sum_i |z_{l,i}^n|$, Rank Regularizer (RR) is

designed to encourage a lower rank of representations and used during network training. Because the usual definition of rank can be very sensitive to the tiny singular values, we use *stable rank* of activation matrix $\mathbf{Z} = [\mathbf{z}_l^1, \ldots, \mathbf{z}_l^{N_{MB}}]^T$ as a surrogate. Note that $N_{MB}$ instead of $N$ activation vectors are used for each mini-batch. Stable rank of $\mathbf{Z}$ is defined as

$$\Omega_{RR} = \frac{\|\mathbf{Z}\|_F^2}{\|\mathbf{Z}\|_2^2} = \frac{\sum_i s_i^2}{\max_i s_i^2}, \tag{7}$$

where $\|\mathbf{Z}\|_F$ is the Forbenius norm, $\|\mathbf{Z}\|_2$ is the spectral norm, and $\{s_i\}$ are the singular values of $\mathbf{Z}$. From $\frac{\sum_i s_i^2}{\max_i s_i^2}$, it can be clearly seen that stable rank is upper bounded by the usual rank that counts strictly positive singular values. Because the spectral norm is based on a singular value decomposition, calculating stable rank's derivative for every mini-batch is a computationally heavy operation. To reduce the computational burden, we introduce an approximation using a special case of Holder's inequality.

$$\Omega_{RR} = \frac{\|\mathbf{Z}\|_F^2}{\|\mathbf{Z}\|_2^2} = \frac{\mathrm{trace}(\mathbf{Z}^T \mathbf{Z})}{\|\mathbf{Z}\|_2^2} \tag{8}$$

$$\geq \frac{\mathrm{trace}(\mathbf{Z}^T \mathbf{Z})}{\|\mathbf{Z}\|_1 \|\mathbf{Z}\|_\infty} = \frac{\sum_{i,n}(z_i^n)^2}{(\max_i \sum_{n=1}^{N_{MB}} |z_i^n|)(\max_n \sum_{i=1}^{M} |z_i^n|)} \tag{9}$$

The inequality $\|\mathbf{Z}\|_2 \leq \sqrt{\|\mathbf{Z}\|_1 \|\mathbf{Z}\|_\infty}$ was used where $\|\mathbf{Z}\|_1$ is the maximum absolute column sum of the matrix $\mathbf{Z}$ (sum of all activation values of unit $i$) and $\|\mathbf{Z}\|_\infty$ is the maximum absolute row sum of the matrix $\mathbf{Z}$ (sum of all activation values of sample $n$). Then the gradient of approximation $\Omega_{RR}$ can be as below.

$$\frac{\partial \Omega_{RR}}{\partial z_i^n} \simeq \frac{\frac{\partial \|\mathbf{Z}\|_F^2}{\partial z_i^n}}{\|\mathbf{Z}\|_1 \|\mathbf{Z}\|_\infty} - \frac{\|\mathbf{Z}\|_F^2 \cdot (\frac{\partial \|\mathbf{Z}\|_1}{\partial z_i^n} \cdot \|\mathbf{Z}\|_\infty + \|\mathbf{Z}\|_1 \cdot \frac{\partial \|\mathbf{Z}\|_\infty}{\partial z_i^n})}{\|\mathbf{Z}\|_1^2 \|\mathbf{Z}\|_\infty^2}, \tag{10}$$

$$\text{where } \frac{\partial \|\mathbf{Z}\|_F^2}{\partial z_i^n} = 2z_i^n,$$

$$\frac{\partial \|\mathbf{Z}\|_1}{\partial z_i^n} = \mathbb{1}_{(i=i^*)} \cdot \mathrm{sign}(z_i^n), \frac{\partial \|\mathbf{Z}\|_\infty}{\partial z_i^n} = \mathbb{1}_{(n=n^*)} \cdot \mathrm{sign}(z_i^n),$$

$$i^* = \arg\max_{1 \leq i \leq M} \sum_{n=1}^{N_{MB}} |z_i^n|, \text{ and } n^* = \arg\max_{1 \leq n \leq N_{MB}} \sum_{i=1}^{M} |z_i^n|.$$

### 4.3 A CONTROLLED EXPERIMENT ON DATA GENERATION PROCESS

We have designed two datasets where the number of independent factors is fully controlled. The first dataset is a synthetic 10-class classification dataset that was created using Python scikit-learn library (Pedregosa et al., 2011). The number of independent Gaussian factors, $d$, was controlled to be 10, 50, 100, 250, and 500, and the independent factors were mixed using a randomly generated $1000 \times d$ rotation matrix. The second dataset is a PCA-controlled MNIST data that was created by including only the top 10, 50, 100, 250, and 500 dimensions of MNIST's PCA dimensions.

For the two datasets, we have chosen $\mathcal{N}_A$ to be the same 5-layer MLP as before and repeatedly performed training while applying either L1R or RR with different loss weights. The results are shown in Figure 2 and Figure 3. The sparsity and rank plots show that indeed sparsity is increased and rank is reduced by increasing the loss weight. The accuracy performance, however, does not show any meaningful dependency on $d$. For instance, even when $d = 10$ and there were only ten independent factors in the data generation process, strongly applying L1R or RR did not result in improved performance. On the contrary, the accuracy often suffered when loss weight was increased.

According to the discussion in subsection 4.1, it is not surprising that the level of learned representation's sparsity does not affect the accuracy performance. Perhaps it is more surprising that even the level of learned representation's rank does not affect the accuracy performance. We move on to the analysis of mutual information for a further discussion on this issue.

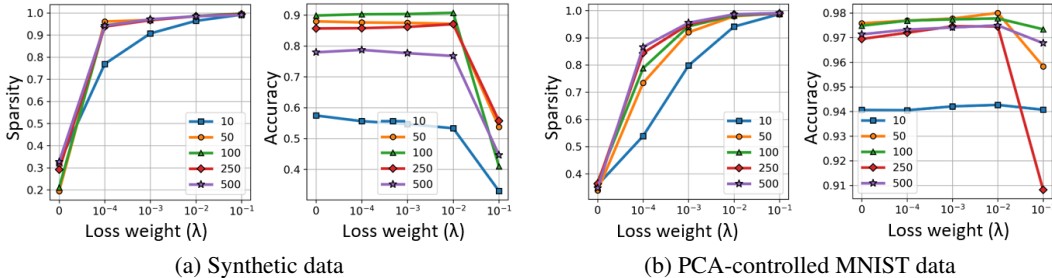

(a) Synthetic data      (b) PCA-controlled MNIST data

Figure 2: Effect of L1R (L1 Representation Regularizer): representation sparsity ($P_s$) and accuracy are shown as a function of L1R's loss weight. Each line corresponds to a different number of independent factors. While sparsity is well controlled, accuracy does not show any meaningful dependency on the number of independent factors used in the data generation process.

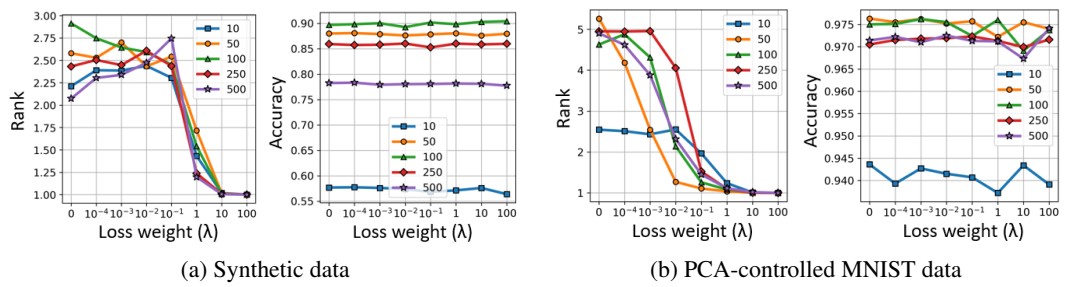

(a) Synthetic data      (b) PCA-controlled MNIST data

Figure 3: Effect of RR (Rank Regularizer): representation rank ($r$) and accuracy are shown as a function of RR's loss weight. Each line corresponds to a different number of independent factors. While rank is well controlled, accuracy does not show any meaningful dependency on the number of independent factors used in the data generation process.

## 5 MUTUAL INFORMATION

So far, we have investigated popular statistical characteristics of representation $\mathbf{z}_l$ where none of the statistical characteristics showed a strong and clear relationship to deep network's performance. In this section, we examine two information-theoretical characteristics: $I(\mathbf{z}_l; x)$ and $I(\mathbf{z}_l; y)$. In the original and pioneering work of Shwartz-Ziv & Tishby (2017), the two characteristics of $\mathbf{z}_l$ were used to explain the concept of information bottleneck on deep networks. Basically, the work shows that the task-relevant information should be maximized via $I(\mathbf{z}_l; y)$ while the task-irrelevant information should be minimized via $I(\mathbf{z}_l; x)$. A further development was made in Achille & Soatto (2017), where the Information Bottleneck Lagrangian $\mathcal{L}(p(\mathbf{z}_l \mid \mathbf{x})) = H(\mathbf{y} \mid \mathbf{z}_l) + \beta I(\mathbf{z}_l; \mathbf{x})$ was explained - the first term is the usual cross entropy cost function, the second term is a penalty term on $I(\mathbf{z}_l; x)$, and $\beta$ is parameter for controlling a trade-off between sufficiency (the first term) and minimality (the second term). In Achille & Soatto (2018), they develop 'information dropout' method that implicitly minimize $I(\mathbf{z}_l; \mathbf{x})$. In their limited performance experiments, they showed that information dropout can improve MNIST classification performance by about 0.25% for the best case. In another work by Kolchinsky et al. (2017), an upper bound derived using a non-parametric estimator of mutual information and a variational approximation is used to develop a gradient-based optimization method. They showed $I(\mathbf{z}_l; \mathbf{x})$ and $I(\mathbf{z}_l; \mathbf{y})$ are indeed reduced by the method, but did not report anything about performance.

In this work, we neither tried the aforementioned techniques nor explicitly implemented an $I(\mathbf{z}_l; \mathbf{x})$ regularizer. Instead, we simply applied the twelve regularizers (including baseline) and calculated the upper and lower bounds of $I(\mathbf{z}_l; \mathbf{x})$ and $I(\mathbf{z}_l; \mathbf{y})$. The bounds can be calculated using the results of Kolchinsky & Tracey (2017) where a pairwise distance function between mixture components is used. They prove that the Chernoff $\alpha$-divergence and the Kullback-Leibler divergence provide lower and upper bounds when they are chosen as the distance function, respectively. Figure 4 shows the results for the last hidden layer together with the generalization error where the same network and dataset as in Figure 1 were used, and regularizers were also applied to the last hidden layer. One

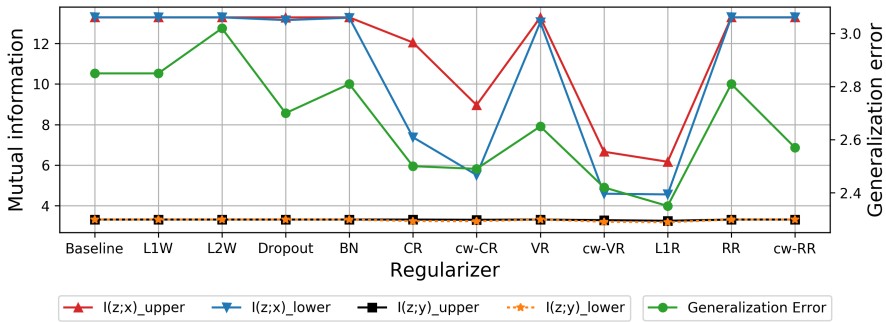

Figure 4: Mutual information and generalization error

can observe that all the regularizers end up with almost the same $I(\mathbf{z}_l; \mathbf{y})$ value. But the bounds of $I(\mathbf{z}_l; \mathbf{x})$ can be seen to be strongly dependent on which regularizer is used, and the upper and lower bounds show a similar pattern as the generalization error's pattern. In fact, the correlation between the lower bound and generalization error can be calculated to be 0.84, and the correlation between the upper bound and generalization error can be calculated to be 0.78. Therefore, it can be surmised that the regularizers might be indirectly affecting the performance by influencing $I(\mathbf{z}_l; \mathbf{x})$.

When a mutual information regularizer is excluded, the rest of the representation regularizers fail to provide general reasoning on why any of the statistical characteristics should be pursued. In fact, one can argue that even a single-neuron in layer $l$ (activation becomes a scalar) can be a sufficient condition for encoding to have a chance to achieve the maximum possible $I(\mathbf{z}_l; \mathbf{y})$, i.e., lossless in terms of relevant information. Such an encoding on a scalar activation might be very inefficient, and a practical learning method might never reach such an encoding. Nonetheless, there is no reason why such an encoding should be impossible. Obviously, many of the statistical characteristics become meaningless for such a scalar representation, and it is high time to reconsider the so-called conventional wisdom on representation characteristics.

## 6 IMPROVING PERFORMANCE WITH REPRESENTATION REGULARIZERS

In the previous sections, we have investigated representation characteristics and their relationship to the performance. All the results, except for mutual information that is shown in Figure 4, indicate that there might be no firm ground to believe that $\mathbf{z}_l$'s representation characteristics are strongly related to the performance. But there have been numerous reports that performance was improved by utilizing newly designed regularizers. In this section, we investigate if (representation) regularizers can indeed consistently improve the performance for a given task condition. Here, a task condition means a learning task with small data size, a small layer width, a specific dataset, a large number of classes, or a specific optimizer. We perform experiments on MNIST, CIFAR-10, and CIFAR-100 datasets using twelve regularizers, and the representation regularizers are explained in Appendix B. The details of experimental settings and architecture parameters can be found in Appendix D.

### 6.1 CONSISTENTLY WELL PERFORMING REGULARIZER

We analyze if there is a consistent dependency between a regularizer and its effect on the performance when a particular regularizer is applied to a particular task condition. Our results, as shown in Table 4, Table 8, and Table 9 indicate that there is no consistently well-performing regularizer for a specific task condition. As an example, consider the entire CIFAR-10 dataset results in Table 4. While task conditions change over different columns, the data remains common for all the tasks. If there is a representation characteristic that fits the data-generation process well and one of the regularizers can match the representation well, it might have outperformed across all the columns. In the table, the best performing regularizer for each task (column) is marked in bold, and any other regularizer whose performance overlaps with the best one is highlighted in green. Looking at the bold and green-highlight patterns, one can easily conclude that there is no single regularizer that works well for all the tasks of CIFAR-10. A similar observation can be made for other task conditions. For instance, one can examine the data size of 1k. For the 1k columns of the three tables, there is no single regularizer that always performs distinctively well.

Table 4: Condition experiment results for CIFAR-10 CNN model. The best performing regularizer in each condition (each column) is shown in bold, and other regularizers whose performance range overlaps with the best one are highlighted in green. For the default condition, the standard values of data size=50k and layer width=128 were used, and Adam optimizer was applied. For the other columns, all the conditions were the same as the default except for the condition indicated on the top part of the columns. Regularizers were applied to the fully-connected layer.

| Regularizer | Default | Data Size | | Layer Width | | Optimizer | |
| --- | --- | --- | --- | --- | --- | --- | --- |
| | | 1k | 5k | 32 | 512 | Momentum | RMSProp |
| Baseline | $26.64 \pm 0.16$ | $56.07 \pm 0.36$ | $43.95 \pm 0.43$ | $28.54 \pm 0.63$ | $28.52 \pm 1.06$ | $25.78 \pm 0.37$ | $28.52 \pm 1.21$ |
| L1W | $26.46 \pm 0.39$ | $56.64 \pm 0.91$ | $44.32 \pm 0.66$ | $28.65 \pm 1.14$ | $27.96 \pm 0.72$ | $25.73 \pm 0.40$ | $28.30 \pm 0.99$ |
| L2W | $25.71 \pm 0.98$ | $56.57 \pm 0.22$ | $44.87 \pm 0.81$ | $28.54 \pm 0.30$ | $27.79 \pm 0.83$ | $26.35 \pm 0.54$ | $28.02 \pm 0.88$ |
| Dropout | $26.38 \pm 0.21$ | $56.11 \pm 0.83$ | $44.78 \pm 0.41$ | $27.66 \pm 0.51$ | $28.43 \pm 0.88$ | $25.95 \pm 0.57$ | $27.69 \pm 0.38$ |
| BN | $31.97 \pm 3.10$ | $56.49 \pm 0.32$ | $43.75 \pm 0.76$ | $28.83 \pm 0.47$ | $28.20 \pm 0.40$ | $25.50 \pm 0.55$ | $28.38 \pm 0.86$ |
| CR | $24.96 \pm 0.63$ | $57.40 \pm 2.11$ | $45.16 \pm 0.94$ | $26.45 \pm 0.22$ | $28.65 \pm 1.21$ | $26.72 \pm 0.61$ | $27.94 \pm 0.43$ |
| cw-CR | $22.99 \pm 0.58$ | $53.50 \pm 1.05$ | $42.15 \pm 0.64$ | $26.40 \pm 0.62$ | $28.54 \pm 1.01$ | $25.93 \pm 0.59$ | $27.77 \pm 0.88$ |
| VR | $21.44 \pm 0.88$ | $53.90 \pm 0.97$ | $42.33 \pm 0.57$ | $\mathbf{24.96 \pm 0.26}$ | $26.61 \pm 0.47$ | $25.01 \pm 0.41$ | $26.06 \pm 0.72$ |
| cw-VR | $21.58 \pm 0.21$ | $\mathbf{51.93 \pm 1.09}$ | $43.00 \pm 0.95$ | $25.81 \pm 0.64$ | $\mathbf{26.46 \pm 0.25}$ | $24.42 \pm 0.31$ | $26.19 \pm 1.35$ |
| L1R | $\mathbf{20.63 \pm 0.50}$ | $52.39 \pm 0.99$ | $\mathbf{40.92 \pm 0.33}$ | $25.49 \pm 0.61$ | $27.81 \pm 0.43$ | $25.13 \pm 0.52$ | $26.49 \pm 0.96$ |
| RR | $26.46 \pm 0.25$ | $57.09 \pm 1.08$ | $44.35 \pm 1.09$ | $26.58 \pm 0.66$ | $26.87 \pm 0.58$ | $\mathbf{23.92 \pm 0.37}$ | $\mathbf{25.80 \pm 0.85}$ |
| cw-RR | $26.29 \pm 0.41$ | $57.55 \pm 0.46$ | $44.71 \pm 1.59$ | $26.62 \pm 0.77$ | $27.12 \pm 0.46$ | $24.34 \pm 0.27$ | $26.10 \pm 0.59$ |
| Best improvement | 6.01 | 4.14 | 3.03 | 3.58 | 2.06 | 1.86 | 2.72 |

In fact, we have experimented many more settings than what are shown in this paper. The hope was to find a strong match between a task condition and a representation regularizer, but we have failed to find anything that looks consistent. Many of the previous works on regularizers have compared their regularizer with only a small number of known regularizers. When many regularizers are compared over many different tasks as in our work, one can easily conclude that there is no obvious relation to declare where a certain representation characteristic is advantageous.

## 6.2 PERFORMANCE IMPROVEMENT USING REGULARIZERS AS A SET

Even though no single representation characteristics consistently outperforms, it can be seen that one can improve performance by using the twelve regularizers as a set and by choosing the best performing regularizer for the given task. This is in line with the usual theme of *tuning* in many areas of deep learning. Looking into Table 4 more carefully, we can see that cw-VR and L1R often had the best performance for CIFAR-10 test cases. In our experiments, we have observed that one of representation regularizers often outperform weight regularizers (L1W, L2W), dropout, and BN. Even though representation regularizers do not seem to have a direct impact on the performance, they might have indirect effects on mutual information as we have seen in Section 5 or on the optimization process. When many representation regularizers are tried as a set, perhaps there is a larger chance of one of such indirect effects improving the performance.

## 7 DISCUSSION AND CONCLUSION

We have studied the most popular statistical characteristics and information theoretic characteristics of DNN representations. All the statistical characteristics that were studied failed to show any general or causal pattern for improving performance. Some of the conventional wisdom were analytically dismissed. Empirical results consistently showed that none of the studied statistical characteristics is a requirement for achieving good performance. While we could not identify any systematic pattern, the popular regularizers have been frequently observed to provide a healthy performance improvement. To understand this phenomenon, we have tried applying twelve different regularizers over many classification tasks with different task conditions. The results show that still no systematic pattern can be found, but the set of regularizers can be used as a very compelling tool for tuning the performance. In contrast to the statistical characteristics, information theoretic characteristic $I(\mathbf{z}_l; \mathbf{x})$ showed a strong correlation with the performance of a classification task. Regularizers were able to affect the mutual information, and possibly they ended up affecting the performance as well. In this work, we have directly addressed and dismissed several conventional wisdom. However, perhaps the most important contribution of this work is to provide an early work on developing rigorous and general theories and methodologies that can be used to better understand the learned representations.

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

# APPENDIX

## A    PROOFS

**Theorem 1** *For a deep network $\mathcal{N}_\mathcal{A} = (\mathbf{W}, \mathbf{b})$ whose layer $l$ is linear, there exists $\widetilde{\mathcal{N}}_\mathcal{A} = (\widetilde{\mathbf{W}}, \widetilde{\mathbf{b}})$ that satisfy the following conditions:*

$$\forall \mathbf{x}, \ \mathcal{N}_\mathcal{A}(\mathbf{x}) = \widetilde{\mathcal{N}}_\mathcal{A}(\mathbf{x});$$
$$\forall \mathbf{x}, \ \widetilde{\mathbf{z}}_l = \mathbf{Q}(\mathbf{z}_l - \mathbf{m}),$$

*where $\mathbf{Q}$ is any nonsingular square matrix of a proper size and $\mathbf{m}$ is any vector of a proper size.*

*Proof.* Proof is based on a simple construction. Choose the weights of $\widetilde{\mathcal{N}}_\mathcal{A}$ as below.

$$\widetilde{\mathbf{W}}_l^T = \mathbf{Q}\,\mathbf{W}_l^T$$
$$\widetilde{\mathbf{b}}_l = \mathbf{Q}(\mathbf{b}_l - \mathbf{m})$$
$$\widetilde{\mathbf{W}}_{l+1}^T = \mathbf{W}_{l+1}^T\,\mathbf{Q}^{-1}$$
$$\widetilde{\mathbf{b}}_{l+1} = \mathbf{b}_{l+1} + \mathbf{W}_{l+1}^T\,\mathbf{m}$$

For all the other layers, choose the same as $\mathcal{N}_\mathcal{A}$'s weights. Then, clearly $\widetilde{\mathbf{z}}_{l-1} = \mathbf{z}_{l-1}$ and therefore $\widetilde{\mathbf{z}}_l = \widetilde{\mathbf{W}}_l^T\,\mathbf{z}_{l-1} + \widetilde{\mathbf{b}}_l = \mathbf{Q}(\mathbf{W}_l^T\,\mathbf{z}_{l-1} + \mathbf{b}_l - \mathbf{m}) = \mathbf{Q}(\mathbf{z}_l - \mathbf{m})$. Also, $\widetilde{\mathbf{z}}_{l+1} = \widetilde{\mathbf{W}}_{l+1}^T\widetilde{\mathbf{z}}_l + \widetilde{\mathbf{b}}_{l+1} = \mathbf{W}_{l+1}^T\,\mathbf{Q}^{-1}\,\mathbf{Q}(\mathbf{z}_l - \mathbf{m}) + \mathbf{b}_{l+1} + \mathbf{W}_{l+1}^T\,\mathbf{m} = \mathbf{z}_{l+1}$. Because the activation vector of layer $l + 1$ is the same for $\mathcal{N}_\mathcal{A}$ and $\widetilde{\mathcal{N}}_\mathcal{A}$, the resulting outputs $\mathcal{N}_\mathcal{A}(\mathbf{x})$ and $\widetilde{\mathcal{N}}_\mathcal{A}(\mathbf{x})$ are exactly the same as well.    $\square$

**Theorem 2** *For a deep network $\mathcal{N}_\mathcal{A} = (\mathbf{W}, \mathbf{b})$ whose activation function of layer $l$ is ReLU, there exists $\widetilde{\mathcal{N}}_\mathcal{A} = (\widetilde{\mathbf{W}}, \widetilde{\mathbf{b}})$ that satisfy the following conditions:*

$$\forall \mathbf{x}, \ \mathcal{N}_\mathcal{A}(\mathbf{x}) = \widetilde{\mathcal{N}}_\mathcal{A}(\mathbf{x});$$
$$\forall \mathbf{x}, \ \widetilde{\mathbf{z}}_l = \mathbf{Q}\,\mathbf{z}_l,$$

*where $\mathbf{Q}$ is any permuted positive diagonal matrix of a proper size. Furthermore, it can be shown that any $\mathbf{Q}$ that satisfy the above two conditions must be a permuted positive diagonal matrix.*

*Proof.* For simplicity, we denote $\mathbf{a}^+ = \mathrm{ReLU}(\mathbf{a})$ and $\mathbf{a}^- = \mathrm{ReLU}(-\mathbf{a})$. Then $\mathbf{a} = \mathbf{a}^+ + \mathbf{a}^-$. We denote $\mathbf{h}_l = \mathbf{W}_l^T\,\mathbf{z}_l + \mathbf{b}_l$, which is the representation before applying the activation function, such that $\mathbf{z}_l = \mathbf{h}_l^+$. If we choose $\widetilde{\mathbf{W}}_l^T = \mathbf{Q}\,\mathbf{W}_l^T$ and $\widetilde{\mathbf{W}}_{l+1}^T = \mathbf{W}_{l+1}^T\,\mathbf{Q}^{-1}$, our focus is to find an invertible matrix $\mathbf{Q}$ that satisfy $\mathbf{W}_{l+1}^T\,\mathbf{Q}^{-1}(\mathbf{Q}\,\mathbf{h}_l)^+ + \mathbf{b}_{l+1} = \mathbf{W}_{l+1}^T\,\mathbf{h}_l^+ + \mathbf{b}_{l+1}$ for all $\mathbf{x}$. This reduces down to finding $\mathbf{Q}$ that satisfies $(\mathbf{Q}\,\mathbf{h}_l)^+ = \mathbf{Q}\,\mathbf{h}_l^+$. We denote $i$'th row of $\mathbf{Q}$ as $\mathbf{q}_i^T$, and the statement mentioned above can be written as: $(\mathbf{q}_i^T\,\mathbf{h}_l)^+ = \mathbf{q}_i^T\,\mathbf{h}_l$. For $\mathbf{q}_i^T$ that satisfy $(\mathbf{q}_i^T\,\mathbf{h}_l) \geq 0$, obviously $(\mathbf{q}_i^T\,\mathbf{h}_l)^+ = \mathbf{q}_i^T\,\mathbf{h}_l^i$. If we substitute $\mathbf{h}_l$ with $\mathbf{h}_l^+ - \mathbf{h}_l^-$, then $\mathbf{q}_i^T\,\mathbf{h}_l^- = 0$. For $\mathbf{q}_i^T$ that satisfy $\mathbf{q}_i^T\,\mathbf{h}_l < 0$, we can derive the condition $\mathbf{q}_i^T\,\mathbf{h}_l^+ = 0$.

Now, we will show that $\mathbf{Q}$ should be a permuted positive diagonal matrix using the statements proved above. For a permuted positive diagonal matrix, $\{\mathbf{q}_i^T\}$ are linearly independent and each $\mathbf{q}_i^T$ has only one positive element. Because $\mathbf{Q}$ needs to be invertible (otherwise information loss occurs and $\mathcal{N}_\mathcal{A}(\mathbf{x}) = \widetilde{\mathcal{N}}_\mathcal{A}(\mathbf{x})$ cannot be achieved), it is trivial that each $\mathbf{q}_i^T$ is linearly independent. To show each $\mathbf{q}_i^T$ has only one positive element, let's assume $\mathbf{q}_i^T$ has more than one non-zero elements. If we denote $\mathbf{q}_{ik}^T$ as the $\mathbf{q}_i^T$'s k-th element, and $\mathbf{h}_{lk}$ as the $\mathbf{h}_l$'s k-th element, we can divide the element indexes as follow.

$$A = \{k|\,\mathbf{q}_{ik}^T \neq 0 \text{ and } \mathbf{h}_{lk} > 0\}$$
$$B = \{k|\,\mathbf{q}_{ik}^T \neq 0 \text{ and } \mathbf{h}_{lk} < 0\}$$

Then, we can consider $\mathbf{h}_l$ such that $A \neq \emptyset$ and $B \neq \emptyset$. If $\mathbf{q}_i^T \mathbf{h}_l >= 0$, then

$$\sum_{j \in A} \mathbf{q}_{ij}^T \mathbf{h}_{lj} > \sum_{j \in B} \mathbf{q}_{ij}^T \mathbf{h}_{lj} .$$

In this case, the right side should be zero because $\mathbf{q}_i^T \mathbf{h}_l^- = 0$. However, $\mathbf{q}_{ij}^T$ should be zero to satisfy $\mathbf{q}_i^T \mathbf{h}_l^- = 0$ because $\mathbf{h}_l$ can be chosen arbitrary in the range that $A \neq \emptyset$ and $B \neq \emptyset$ and $\mathbf{q}_i^T \mathbf{h}_l >= 0$. This is a contradiction due to the definition of B. We can prove the case of $\mathbf{q}_i^T \mathbf{h}_l < 0$ similarly, which shows that each $\mathbf{q}_i^T$ has only one non-zero element. To show the $\mathbf{q}_i^T$'s one element is positive, we denotes the $\mathbf{q}_i^T$'s only one element as $\mathbf{q}_{ij}^T$, in which $j$ is the index of the non-zero element. When $\mathbf{q}_{ij}^T < 0$, we can consider $\mathbf{h}_l$ such that $\mathbf{h}_{lj} < 0$. In this case, $(\mathbf{q}_i^T \mathbf{h}_l) > 0$, but $\mathbf{q}_i^T \mathbf{h}_l^- > 0$, which contradicts the condition $\mathbf{q}_i^T \mathbf{h}_l^- = 0$. If we consider $\mathbf{h}_l$ such that $\mathbf{h}_{lj} > 0$, $(\mathbf{q}_i^T \mathbf{h}_l) < 0$, but $\mathbf{q}_i^T \mathbf{h}_l^+ > 0$, which contradicts the condition $\mathbf{q}_i^T \mathbf{h}_l^+ > 0$. Therefore, $\mathbf{q}_{ij}^T$ is positive. $\qquad \square$

## B    REPRESENTATION REGULARIZERS

In this section, we briefly introduce the representation regularizers that are used in our experiments. Based on the notations provided in Section 2, we define class-wise statistics that are calculated using only class $k$'s samples out of a total of $K$ labels in the mini-batch. Class-wise mean, covariance, and variance are defined as below.

$$\mu_{\mathbf{z}_l,i}^{(k)} = \mathbb{E}_{n \in S_k}[z_{l,i}^n]. \tag{11}$$

$$c_{i,j}^{(k)} = \mathbb{E}_{n \in S_k}[(z_{l,i}^n - \mu_{\mathbf{z}_l,i}^{(k)})(z_{l,j}^n - \mu_{\mathbf{z}_l,j}^{(k)})]. \tag{12}$$

$$v_{\mathbf{z}_l,i}^{(k)} = c_{i,i}^{(k)}. \tag{13}$$

Here, $S_k$ is the set that contains indexes of the samples with class label $k$. Note that superscripts with and without parenthesis indicate class label and sample index, respectively. Penalty loss functions of the representation regularizers are summarized in Table 5.

Table 5: Penalty loss functions of representation regularizers.

| Penalty loss function | Description on regularization term |
| --- | --- |
| $\Omega_{CR} = \sum_{i \neq j} (c_{i,j})^2$ | *Covariance* of representations calculated from all-class samples. |
| $\Omega_{cw\text{-}CR} = \sum_{k} \sum_{i \neq j} (c_{i,j}^{(k)})^2$ | *Covariance* of representations calculated from **the same class samples**. |
| $\Omega_{VR} = \sum_{i} v_{\mathbf{z}_l,i}$ | *Variance* of representations calculated from all-class samples. |
| $\Omega_{cw\text{-}VR} = \sum_{k} \sum_{i} v_{\mathbf{z}_l,i}^{(k)}$ | *Variance* of representations calculated from **the same class samples**. |
| $\Omega_{L1R} = \sum_{n} \sum_{i} | z_{l,i}^n |$ | *Absolute amplitude* of representations calculated from all-class samples. |
| $\Omega_{RR} = \dfrac{\|\mathbf{Z}_l\|_F^2}{\|\mathbf{Z}_l\|_2^2}$ | *Stable rank* of representations calculated from all-class samples. |
| $\Omega_{cw\text{-}RR} = \sum_{k} \dfrac{\left\|\mathbf{Z}_l^{(k)}\right\|_F^2}{\left\|\mathbf{Z}_l^{(k)}\right\|_2^2}$ | *Stable rank* of representations calculated from **the same class samples**. |

## C    REPRESENTATION CHARACTERISTICS

In this section, we investigate the statistical characteristics of the learned representations when different regularizers are applied. We used the same network and dataset as the ones used for

generating Figure 1. All the regularizers were applied only to the fifth layer, and the representation characteristics were calculated using the fifth layer as well. In Table 6, typical evaluation results of statistical characteristics are shown. We can confirm that the statistical characteristics targeted by each representation regularizer are indeed manipulated as expected (**Bold**). In particular, RR and cw-RR designed in this work to regularize the stable rank work as expected. The two weight regularizers (L1W: L1 Weight Regularizer, L2W: L2 Weight Regularizer) have similar characteristic values as the baseline's, and this can be taken for granted because the regularizers do not directly regularize representations. A few conventional beliefs mentioned in the paper are quantitatively confirmed as well. A large number of dead units is known to be harmful because they do not contribute toward improving the performance of deep networks. Our result shows even 39% of DEAD UNIT caused by L1R does not hurt the performance, which is in line with our analysis in Section 4. For dropout, COVARIANCE is reduced as in Cogswell et al. (2015), but CORRELATION is actually increased compared to the baseline. In fact, COVARIANCE is reduced simply because ACTIVATION AMPLITUDE is reduced as mentioned in Section 3, and the correlation between two active units is actually made larger by applying dropout. Therefore, it cannot be said that the relationship between a pair of neurons becomes weaker by applying dropout. This is in contrary to the 'reduction of co-adaptation' idea. Note that we have excluded the inactive neurons for the evaluations. If the inactive ones are included with their zero values, the covariance and correlation values will be different.

Table 6: Statistical characteristics of learned representations.

| Regularizer | Error | ACTIVATION AMPLITUDE | COVARIANCE | CORRELATION | SPARSITY | DEAD UNIT | RANK |
|---|---|---|---|---|---|---|---|
| Baseline | 2.85 | 4.93 | 2.08 | 0.27 | 0.34 | 0.13 | 2.41 |
| L1W | 2.85 | 4.53 | 1.95 | 0.28 | 0.29 | 0.01 | 2.32 |
| L2W | 3.02 | 4.76 | 2.23 | 0.29 | 0.34 | 0.09 | 2.26 |
| Dropout | 2.70 | 2.72 | 0.87 | *0.42* | 0.58 | 0.06 | 2.75 |
| BN | 2.81 | 1.35 | 0.24 | 0.28 | 0.52 | 0.00 | 5.14 |
| CR | 2.50 | 0.50 | **0.01** | **0.19** | 0.40 | 0.03 | 7.12 |
| cw-CR | 2.49 | 0.63 | 0.02 | 0.31 | 0.51 | 0.07 | 3.60 |
| VR | 2.65 | 1.35 | 0.15 | 0.26 | 0.40 | 0.08 | 3.92 |
| cw-VR | 2.42 | 0.63 | 0.02 | 0.36 | 0.53 | 0.06 | 3.90 |
| L1R | 2.35 | 1.29 | 0.03 | 0.40 | **0.97** | *0.39* | 5.94 |
| RR | 2.81 | 7.23 | 226.2 | 0.90 | 0.43 | 0.18 | **1.00** |
| cw-RR | 2.57 | 10.31 | 96.3 | 0.91 | 0.31 | 0.22 | **1.00** |

## D  EXPERIMENT DETAILS

### D.1  DEFAULT SETTINGS

By default, ReLU activation function and Adam optimizer were used, and a learning rate was set to 0.0001. Validation performance was evaluated with different loss weights {0.001, 0.01, 0.1, 1, 10, 100}, and the one with the best validation performance for each regularizer and condition was chosen for testing. We used the validation data for MNIST dataset and the last 10,000 samples of training data for CIFAR-10/100 datasets as validation data. After a loss weight was fixed, the validation data was merged back into the training data in the case of CIFAR-10/100. Five training trials were performed, and the means of the five trials were reported. Note that a mini-batch size was set to 100 for MNIST and CIFAR-10 tasks but to 500 for CIFAR-100 due to the class-wise statistic calculation. In this work, we carried out all the experiments using TensorFlow 1.5.

### D.2  ARCHITECTURE PARAMETERS

We used a 6-layer MLP with 100 units per hidden layer for MNIST image classification tasks. We used a CNN with four convolutional layers and one fully-connected layer for CIFAR-10/100 image classification tasks. Architecture details are described in Table 7.

### D.3  EXPERIMENTAL CONDITIONS

Default conditions are shown in bold, and the full experimental conditions are listed below.

- Training data size: 1k, 5k, **50k**
- Layer width: (MNIST) 2, 8, **100** / (CIFAR-10/100): 32, **128**, 512

Table 7: Architecture hyperparameters of CIFAR-10/100 CNN model.

| Layer | Parameter |
|---|---|
| Convolutional layer-1 | Number of filters=32, Filter size=3 × 3, Convolution stride=1 |
| Convolutional layer-2 | Number of filters=64, Filter size=3 × 3, Convolution stride=1 |
| Max-pooling layer-1 | Pooling size=2 × 2, Pooling stride=2 |
| Convolutional layer-3 | Number of filters=128, Filter size=3 × 3, Convolution stride=1 |
| Max-pooling layer-2 | Pooling size=2 × 2, Pooling stride=2 |
| Convolutional layer-4 | Number of filters=128, Filter size=3 × 3, Convolution stride=1 |
| Max-pooling layer-3 | Pooling size=2 × 2, Pooling stride=2 |
| Fully connected layer | Number of units=128 |

- Optimizer (CIFAR-10): **Adam**, Momentum (lr=0.01, momentum=0.9), RMSProp (lr=0.0001)
- Number of classes (CIFAR-100): 16, 64, **100**

### D.4 EXPERIMENTAL RESULTS (MNIST, CIFAR-100)

Table 8: Condition experiment results for MNIST MLP model. The best performing regularizer in each condition (each column) is shown in bold, and other regularizers whose performance range overlaps with the best one are highlighted in green. For the default condition, the standard values of data size=50k and layer width=100 were used and Adam optimizer was applied. For other columns, all the conditions were the same as the default except for the condition indicated on the top part of the columns. Regularizers were applied to the fifth hidden layer.

| Regularizer | Default | Data Size | | Layer Width | |
|---|---|---|---|---|---|
| | | 1k | 5k | 2 | 8 |
| Baseline | 2.85 ± 0.11 | 11.41 ± 0.19 | 6.00 ± 0.07 | 31.62 ± 0.07 | 10.52 ± 0.57 |
| L1W | 2.85 ± 0.06 | 11.64 ± 0.27 | 5.96 ± 0.11 | 31.67 ± 0.15 | 11.02 ± 0.58 |
| L2W | 3.02 ± 0.40 | 11.38 ± 0.18 | 5.86 ± 0.10 | 31.66 ± 0.13 | 10.65 ± 0.23 |
| Dropout | 2.70 ± 0.08 | **10.29 ± 0.23** | **5.59 ± 0.11** | 62.09 ± 1.32 | 13.94 ± 1.05 |
| BN | 2.81 ± 0.12 | 10.81 ± 0.04 | 5.60 ± 0.10 | 42.08 ± 0.93 | **7.51 ± 0.58** |
| CR | 2.50 ± 0.05 | 11.63 ± 0.24 | 6.05 ± 0.06 | 34.80 ± 0.25 | 10.25 ± 0.74 |
| cw-CR | 2.49 ± 0.10 | 10.62 ± 0.05 | 5.80 ± 0.15 | 31.50 ± 0.11 | 10.81 ± 1.11 |
| VR | 2.65 ± 0.11 | 14.42 ± 0.14 | 6.90 ± 0.22 | 32.39 ± 0.13 | 9.22 ± 0.28 |
| cw-VR | 2.42 ± 0.06 | 10.44 ± 0.18 | 5.90 ± 0.12 | **30.34 ± 0.06** | 10.01 ± 0.63 |
| L1R | **2.35 ± 0.08** | 11.60 ± 0.20 | 6.20 ± 0.13 | 64.39 ± 0.26 | 88.65 ± 0.00 |
| RR | 2.81 ± 0.10 | 10.92 ± 0.17 | 6.61 ± 0.05 | 38.35 ± 0.20 | 12.31 ± 0.16 |
| cw-RR | 2.57 ± 0.08 | 10.89 ± 0.19 | 6.60 ± 0.17 | 38.57 ± 0.12 | 12.63 ± 0.39 |
| Best improvement | 0.5 | 1.12 | 0.41 | 1.28 | 3.01 |

Table 9: Condition experiment results for CIFAR-100 CNN model. The best performing regularizer in each condition (each column) is shown in bold, and other regularizers whose performance range overlaps with the best one are highlighted in green. For the default condition, the standard values of data size=50k, layer width=128, and number of classes=100 were used. For other columns, all the conditions were the same as the default except for the condition indicated on the top part of the columns. Regularizers were applied to the fully-connected layer.

| Regularizer | Default | Data Size | | Layer Width | | Number of Classes | |
|---|---|---|---|---|---|---|---|
| | | 1k | 5k | 32 | 512 | 16 | 64 |
| Baseline | 61.26 ± 0.52 | 90.89 ± 0.30 | 82.21 ± 0.72 | 62.41 ± 0.34 | 61.30 ± 0.64 | 45.75 ± 0.73 | 58.02 ± 0.40 |
| L1W | 60.97 ± 0.64 | 91.33 ± 0.37 | 82.3 ± 0.6 | 62.23 ± 0.58 | 60.92 ± 0.47 | 45.08 ± 1.53 | 58.08 ± 1.18 |
| L2W | 60.23 ± 0.31 | 90.53 ± 0.39 | 82.05 ± 0.70 | 62.78 ± 0.36 | 61.55 ± 0.99 | 45.28 ± 1.59 | 57.47 ± 0.66 |
| Dropout | 63.88 ± 0.72 | **90.22 ± 0.48** | 81.68 ± 0.81 | 64.08 ± 0.99 | 64.31 ± 0.37 | 45.73 ± 1.57 | 59.14 ± 0.46 |
| BN | 60.93 ± 0.39 | 91.18 ± 0.36 | 82.01 ± 0.58 | 62.18 ± 1.49 | 62.16 ± 0.57 | 44.55 ± 1.43 | 57.72 ± 0.66 |
| CR | 59.88 ± 0.50 | 91.70 ± 0.14 | 82.47 ± 0.41 | **60.47 ± 0.63** | 60.70 ± 0.94 | 44.55 ± 1.10 | 56.76 ± 0.86 |
| cw-CR | 57.03 ± 0.73 | 90.85 ± 0.29 | 81.29 ± 0.62 | 61.41 ± 0.67 | 58.02 ± 0.25 | 43.50 ± 1.21 | 54.24 ± 0.64 |
| VR | 57.68 ± 0.94 | 91.43 ± 0.32 | 81.85 ± 0.38 | 61.35 ± 0.45 | 56.87 ± 0.74 | 42.33 ± 1.03 | 54.32 ± 0.40 |
| cw-VR | 56.75 ± 0.64 | 90.45 ± 0.22 | **81.03 ± 0.57** | 60.67 ± 0.59 | 56.91 ± 0.73 | **41.38 ± 0.53** | 54.23 ± 1.06 |
| L1R | **56.03 ± 0.81** | 91.15 ± 0.35 | 81.98 ± 0.47 | 61.11 ± 0.31 | **56.46 ± 0.62** | 42.51 ± 1.43 | **53.65 ± 1.00** |
| RR | 62.68 ± 0.35 | 91.20 ± 0.27 | 81.32 ± 0.36 | 68.54 ± 0.46 | 59.29 ± 0.32 | 44.16 ± 0.80 | 60.25 ± 0.35 |
| cw-RR | 62.62 ± 0.31 | 90.62 ± 0.34 | 81.57 ± 0.14 | 68.11 ± 0.31 | 59.15 ± 0.29 | 44.10 ± 0.65 | 60.03 ± 0.41 |
| Best improvement | 5.23 | 0.67 | 1.18 | 1.94 | 4.84 | 4.37 | 4.37 |

