# OpenReview forum: "On the Statistical and Information Theoretical Characteristics of DNN Representations"
_ICLR.cc/2019/Conference_

### Official Review · AnonReviewer2 · 2018-10-30
**Interesting ideas, but extremely bold conclusions not rigorously justified**

**Rating:** 3
**Confidence:** 3

**Review:**

This paper investigates the use of techniques for improving neural network training (regularization, normalization of covariance, sparsity) in terms of their generalization properties, empirically and analytically. The claim is that most of these tools do not help improve performance, with the exception of mutual information.

Pros: It's interesting to investigate and compare these different "regularization" techniques and compare them on different tasks empirically.

Cons:
Many of the points made in the paper are not properly capturing the nuance in the "conventional wisdom", and although it's good to be reminded and the empirical results are interesting to look at, in fact these are not really new discoveries, and sometimes the conclusions are very misleading.

1) There is test loss, and there is generalization loss, and it isn't exactly the same thing. For a hypothesis class H, we have

test loss <= train loss + generalization loss

where train loss measures how well we've fit a particular sample, and generalization loss measures how well a model that is trained on one sample can fit a new sample. Note that if I apply L2 loss to my model and have a regularization parameter --> infty, my train loss is huge but my generalization loss is 0. In other words, for a large enough regularization parameter, most of the methods experimented here WILL limit effective capacity and minimize generalization loss; it just will not give you the best test error performance. So the distinction here should be made much clearer--conventional wisdom for regularization limiting generalization error is not wrong.

2) The point that is trying to be made in section 3.1 is somewhat well-known in the general optimization literature. Let's consider a much simpler example: linear regression

min_x ||Ax-b||_2^2 + gamma ||x||_2^2

Let's consider first no regularization, gamma = 0. Assume that A has a huge nullspace. Then technically there are an infinite number of globally optimal solutions x, although if we solve this problem using SGD starting with x = 0, it is known that the minimum norm solution is always picked. You can also think of this as whitening, since large lambda smooths the spectrum of the Hessian. Now add in regularization. Now the solution is unique, even if A is ill-conditioned. It's true that it isn't super necessary to add this regularization, since SGD can get you a good solution, but now we can GUARANTEE that the generalization error is 0. In practice, also, regularization adds stability to the numerics.

In deep learning, the hidden layer z also acts as a coefficient matrix for determining y. I assume that is why people pursue low correlation, since it affects the conditioning of z.

3) Comments like  "for scaling and permutation, their influences are rather insignificant" seem a bit careless to me. In fact it is well known that scaling can affect training performance significantly. But of course, if I know the global solution for z which feeds into a softmax, then any scaling on z does not affect the output of the softmax. That, however, does not mean I don't care about the scaling of z when training.

4) Sparsity and rank BOTH limit the degrees of freedom. In fact, sparsity makes more sense when optimizing a nonlinear objective, which is always the case in deep learning. The reason to limit rank is when you wish to "learn your codewords", e.g. the eigenvectors, whereas in sparsity, the "codewords" are already learned, and you just learn the weighting. But if the codewords span the space, they both have equal representability.

4) It is not clear to me what the task is in 4.3. What is the "accuracy" in a data generation task? Is this the normal classification task? If so, is the accuracy reported train or test accuracy? How exactly is generalization error being measured?


5)I am not clear as to what conclusion is being drawn in section 5, with the last sentence "obviously, many of the statistical characteristics become meaningless for such a scalar representation, and it is high time to reconsider the so-called conventional wisdom on representation characteristics." why is this conclusion drawn based on the observation that, if a scalar z perfectly correlates with y, in fact this is the most generalization neural network?

6) Table 4: how did you choose your hyperparameters? (regularization performance is extremely sensitive to parameter choice.)

7) A major concern is that basically very little training is done in these comparisons, except in the very last section. As I previously mentioned, many of these regularization / normalization techniques are also meant to better condition the optimization itself, and thus this advantage should not be discarded.


minor comments:
 - page 5 last sentence "characteristics" should be singular
 - page 8 first sentence "to [a] deep network's performance"

---

> ### Author Response · Authors · 2018-11-26
> **Response 1:**
>
> Thanks for your helpful feedback. Overall, we agree with your insightful assessments, and we will need to make proper updates that will take a while. For the sake of discussion, please see our responses below.
> =================
> >>The claim is that most of these tools do not help improve performance, with the exception of mutual information.
> Response: In fact, we have found applying regularizers very helpful for improving DNN performance, especially the newly designed representation regularizers. (For instance, please see https://arxiv.org/abs/1809.09307 for a related work.) What we are claiming in this work is NOT ‘regularizers are *not* helpful’. Rather, we are saying that ‘in many cases, a statistical property should not be quoted as the CAUSE of the performance improvement without providing a sufficient reasoning’. We are not aware of any previous work where this issue is explicitly dealt with.
> =================
> Response to feedback 1)
> >> ``  conventional wisdom for regularization limiting generalization error is not wrong``
> We definitely agree with your explanation on generalization loss. After reading your review, we realized that our writing (and possibly our thinking as well) was not crisp at all. Thanks for reminding us of the statistical learning theory, and we will make a proper adjustment in the future version.
>
> The original intention of the writing was somewhat different. In our work, we were more concerned of how people typically come up with a new regularizer and attempt to explain the performance improvement seen from the limited set of empirical results. It would be easier for us to agree if one says ``optimization conditioning might have contributed toward the improved performance when our newly designed regularizer was used``. Many of the existing works, however, comment on improved generalization due to the new regularizer that affects a statistical property. Our observation was that this type of explanation is quite commonly used or implied in many of the existing works. In your explanation, ``  regularization parameter --> infty``   is an easy and well-known case. But when a regularizer is really used, the regularization parameter is tuned and we believe it is never tuned to make the DNN’s effective capacity to be meaningfully smaller (for instance, Zhang et al.’s ``rethinking generalization``  explains this as you must be familiar with). Therefore, we wanted to make it clear that such statements should not be implied unless they have proper grounds.
>
> =================
> Response to feedback 2)
>
> We understand optimization and conditioning very well, but thanks for the comment at the end. It is good to know that experienced researchers think ``  people pursue low correlation, since it affects the conditioning of z``.  It certainly makes sense to us, and we were not fully aware of the thinking. We also thought the effect of regularizers on optimization would affect the performance, but was not thinking based on conditioning. We simply tried to address it as a ``tuning effect`` as written in section 6. BTW, we were able to improve performance using representation regularizers for most of the tasks that we have tried, and the results are summarized in Section 6 and also in the supplementary. Currently, we are investigating more complicated tasks since the dynamics might not be the same.
>
> =================
> Response to feedback 3)
> Agreed. It was a sloppy writing.
>
> =================
> Response to feedback 4)   (first comment)
> Thanks for the feedback. This is a subtle matter, and at least our empirical test results are more in line with your comment. We will investigate more complicated tasks to better understand this matter.
>
> =================
> Response to feedback 4)   (second comment)
> Normal classification task & test accuracy. Generalization error is estimated as the gap between training error and test error.

---

> > ### Comment · AnonReviewer2 · 2018-11-26
> > **response to rebuttal**
> >
> > Thanks for the response. After reading the rebuttal, I have a better understanding as to what the goal of the paper is, and I agree that, if you can convey the subtleties of your claims, it is a valuable contribution.
> >
> > Probably my biggest suggestion is to separate your analysis on training and generalization more obviously, perhaps by reporting also # iterations until convergence, sensitivity to parameter tuning, etc, as well as generalization error and statistical characterizations. That way it is clear your argument is "why regularizers are good", and not "whether they are good".
> >
> > Additionally, as you mentioned, an extensive evaluation of standard DNNs on a variety of challenging tasks, spanning different input data types (images, structured images, text, audio, etc) as well as standard regularization techniques would make this work extremely interesting, even if the "why" arguments remain mostly empirical; while theorems and proofs are always much more impressive, since they are so hard to do rigorously for DNNs often intuition and extensive experiments are more valuable anyway.
> >
> >
> >
> > >> It would be easier for us to agree if one says ``optimization conditioning might have contributed toward the improved performance when our newly designed regularizer was used``. Many of the existing works, however, comment on improved generalization due to the new regularizer that affects a statistical property. Our observation was that this type of explanation is quite commonly used or implied in many of the existing works. In your explanation, ``  regularization parameter --> infty``   is an easy and well-known case. But when a regularizer is really used, the regularization parameter is tuned and we believe it is never tuned to make the DNN’s effective capacity to be meaningfully smaller (for instance, Zhang et al.’s ``rethinking generalization``  explains this as you must be familiar with). Therefore, we wanted to make it clear that such statements should not be implied unless they have proper grounds.
> >
> > Ok, that is an interesting argument; thanks for the clarification.
> >
> > >>  We also thought the effect of regularizers on optimization would affect the performance, but was not thinking based on conditioning. We simply tried to address it as a ``tuning effect`` as written in section 6.
> >
> > Yeah I would be curious to see how "implicit regularizers" (e.g. dropout and batch normalization) affect numerics. Probably the easiest way to test this is to try a number of hyperparameters / random initialization and measure the sensitivity of the result.
> >
> >
> > >> Normal classification task & test accuracy. Generalization error is estimated as the gap between training error and test error.
> >
> > Thanks for the clarification. Maybe an interesting thing to evaluate as well is sensitivity to aberrant data. Perhaps include a few mislabeled points in the training set, and see if regularization helps ignore these mistakes. Or, model sensitivity; add tiny random perturbations to the final trained model and see if that affects the results. These might be other benefits to generalization that aren't well-conveyed in the standard tasks.
> >
> > >> The sentence was meant to remind of the very extreme case where a single scalar can carry all the information that needs to be passed to the upper layer. While DNN is extremely unlikely to be trained to use such a representation, it would be difficult to prove that it is impossible, either. If such a representation is indeed constructed by DNN, however, almost none of the commonly considered statistical properties would be meaningful. That’s all we are saying.
> >
> > I see what you're saying, and it's an important point. This may be an interesting direction to explore more, though, because if a task can be well-modeled with some complicated representation and 1 degree of freedom, then in the ideal case, proper "regularization" would exactly find this representation in all cases. Something like regularization w.r.t. a data-dependent manifold, where the manifold is also learned... maybe the bottleneck in autoencoders can do something like this?
> >
> > =================
> > >> ``Validation performance was evaluated with different loss weights {0.001, 0.01, 0.1, 1, 10,100}, and the one with the best validation performance for each regularizer and condition was chosen for testing.``
> >
> > Ok, that sounds valid.
> >
> > Overall this is a very interesting and important direction to go down, and I look forward to any future revisions!

---

> > > ### Author Response · Authors · 2018-11-28
> > > **RE**
> > >
> > > Awesome. Good to hear positive feedback. :) We will work on the revision. Thanks.

---

> ### Author Response · Authors · 2018-11-26
> **Response 2:**
>
> Please see our responses below.
> =================
> Response to feedback 5)
> The sentence was meant to remind of the very extreme case where a single scalar can carry all the information that needs to be passed to the upper layer. While DNN is extremely unlikely to be trained to use such a representation, it would be difficult to prove that it is impossible, either. If such a representation is indeed constructed by DNN, however, almost none of the commonly considered statistical properties would be meaningful. That’s all we are saying.
> We agree that the statement is confusing, and will remove it in the future version.
>
> =================
> Response to feedback 6)
> Sorry, the details are in supplementary, Appendix D. ``Validation performance was evaluated with different loss weights {0.001, 0.01, 0.1, 1, 10,100}, and the one with the best validation performance for each regularizer and condition was chosen for testing.``
>
> =================
> Response to feedback 7)
> We did many experiments, but we were also thinking that more experiments on more challenging tasks are needed. While ION results would be applicable anyway, such tasks might lead to different findings on the representation statistics.
>
> =================
> Overall, thanks for your helpful feedback. We truly appreciate your time and comments. (Just in case you have any suggestion for improving this work, please feel free to contact us directly.)

---

### Official Review · AnonReviewer3 · 2018-10-31
**important topic but weak results**

**Rating:** 4
**Confidence:** 4

**Review:**

This paper studies the characteristics of representations and their roles in neural network expressiveness. The results  are overall not very impressive.

1. There are many characteristics of representations such as scaling, permutation, covariance, correlation, sparsity, dead units, rank. The papers discusses some (not surprising) theoretical properties relating to scaling, permutation, covariance, correlation, while making less efforts on the more interesting characteristics  sparsity, dead units, rank, mutual information. Only some heuristic results are obtained for them without rigorous theory. It would be better if these heuristic arguments can be formed as theorems as well.

2. Probably the most interesting experimental finding of this paper is that the mutual information between z and output is constant, while the one between z and input strongly depends on the regularizers. That is, the dependence between z and y does not vary with regularizers but the one between z and x does. Is this a coincidence or a general phenomenon? Is there a theoretical explanation?


3.

---

> ### Author Response · Authors · 2018-11-26
> **Response:**
>
> Thanks for your review. The ION theorem is an important part of explaining sparsity, dead units, and rank as well, but perhaps our writing was not clear enough. We will work on the writing in the future version of this work. As for the mutual information related comment (#2), the results that you have mentioned are well known from information bottleneck paper or from the following information invariance paper.
>
> Alessandro Achille and Stefano Soatto. Emergence of invariance and disentangling in deep representations. Journal of Machine Learning Research. 2018

---

> > ### Comment · AnonReviewer3 · 2018-11-26
> > **Response to rebuttal**
> >
> > Thanks for response.

---

### Official Review · AnonReviewer1 · 2018-11-03
**A preliminary effort to investigate the significance of statistical characteristics of DNN representations**

**Rating:** 5
**Confidence:** 3

**Review:**

This paper makes concerted efforts to examine the existing beliefs about the significance of various statistical characteristics of hidden layer activations (or representations) in a DNN. In the past, many works have argued for encouraging the certain statistical behavior of these representations (e.g., sparsity, low correlation etc) in order to have better classification accuracy. However, this paper tries to argue that such efforts are not very useful as these statistical characteristics don't provide any systematic explanation for the performance of DNNs across different settings.

First, the paper argues that given a DNN, it's possible to construct either an identical output network or a comparable network that can have very different behavior for some of the statistical characteristics. This casts doubt on the usefulness of these characteristics in explaining the performance of the network. The paper conducts experiments with different regularizers associated with some of the standard statistical characteristics using the MNIST, CIFAR-10, and CIFAR-100 datasets. The paper claims that for each dataset the best performing network cannot be attributed to any single regularizer. For the same set of regularizers and the MNIST dataset, the paper then explores the mutual information between the inputs and the hidden layer activations. The paper observes that the best performing regularizer is the one which minimizes this mutual information. Therefore, it is plausible that the mutual information regularization can consistently explain the performance of an NN.

The paper addresses an interesting problem and makes some good contributions. However, the reviewer feels that the brief treatment of mutual information regularizer leaves something to be desired. Did the authors also examine the relationship between mutual information and generalization error for CIFAR data sets? Does it not make sense to examine this for all (most of) the setups considered in Table 4, 8, and 9.

In these tables, how do the authors decide which hidden layer representations should be explored for their statistical characteristics?

The reviewer feels that for CIFAR-10 and 100, some regularizers do consistently give best or close to best networks. Could the authors comment on this?

---

> ### Author Response · Authors · 2018-11-26
> **Response:**
>
> Thank you for your review. We agree that further investigation is needed for mutual information, and we are currently working on it. As for the layer to investigate, we have presented the higher layer results because the representation regularizers showed the most improvements when applied to the higher (or even output) layer. We believe the representations in the lower layers are inherently less structured and therefore representation shaping can be harmful. The layer dependency is further explained in the following article.
>
> Daeyoung Choi and Wonjong Rhee, Utilizing class information for deep network representation shaping, AAAI 2019   (https://arxiv.org/abs/1809.09307)
>
> >> The reviewer feels that for CIFAR-10 and 100, some regularizers do consistently give best or close to best networks. Could the authors comment on this?
> Response: In general, representation regularizers showed better performance than the others. Among the representation regularizers, cw-VR and L1R frequently achieved the best performance. Nonetheless, we were not able to identify any specific task condition that makes a specific regularizer consistently best performing regularizer.

---

> > ### Comment · AnonReviewer1 · 2018-11-30
> > **Response to rebuttal**
> >
> > Thank you for clarifying the point about which layers should be explored for various statistical properties. It's good to know that you are currently investigating mutual information further, which would make the contributions of this paper more interesting.

---

### Meta-Review · Area_Chair1 · 2018-12-13
**Important topic but more work is required**

**Confidence:** 4
**Recommendation:** Reject

**Metareview:**

The paper considers an important problem of investigating the effects different statistical characteristics of representations (hidden unit activations) , such as sparsity, low correlation, etc, have on the neural network performance; while all reviewers agree that this is clearly a very important topic, there is also a consensus that perhaps the authors must strengthen and emphasize their contribution more clearly.